# SMART-Q: An Integrative Pipeline Quantifying Cell Type-Specific RNA Transcription

Xiaoyu Yang[1©], Seth Bergenholtz[1©], Lenka Maliskova[1], Mark-Phillip Pebworth[2,3], Arnold R. Kriegstein[3,4], Yun Li[5], Yin Shen[1,4]*

1 Institute for Human Genetics, University of California San Francisco, San Francisco, CA, United States of America, 2 Biomedical Sciences Graduate Program, University of California San Francisco, San Francisco, CA, United States of America, 3 The Eli and Edythe Broad Center of Regeneration Medicine and Stem Cell Research, UCSF, San Francisco, CA, United States of America, 4 Department of Neurology, University of California San Francisco, San Francisco, CA, United States of America, 5 Department of Genetics, Department of Biostatistics, and Department of Computer Science, University of North Carolina, Chapel Hill, NC, United States of America

© These authors contributed equally to this work.
* yin.shen@ucsf.edu

**Data Availability Statement:** The software files is available from the on Github database (https://github.com/shenlab-ucsf/SMART-Q).

## Abstract

Accurate RNA quantification at the single-cell level is critical for understanding the dynamics of gene expression and regulation across space and time. Single molecule FISH (smFISH), such as RNAscope, provides spatial and quantitative measurements of individual transcripts, therefore, can be used to explore differential gene expression among a heterogeneous cell population if combined with cell identify information. However, such analysis is not straightforward, and existing image analysis pipelines cannot integrate both RNA transcripts and cellular staining information to automatically output cell type-specific gene expression. We developed an efficient and customizable analysis method, Single-Molecule Automatic RNA Transcription Quantification (SMART-Q), to enable the analysis of gene transcripts in a cell type-specific manner. SMART-Q efficiently infers cell identity information from multiplexed immuno-staining and quantifies cell type-specific transcripts using a 3D Gaussian fitting algorithm. Furthermore, we have optimized SMART-Q for user experiences, such as flexible parameters specification, batch data outputs, and visualization of analysis results. SMART-Q meets the demands for efficient quantification of single-molecule RNA and can be widely used for cell type-specific RNA transcript analysis.

## Introduction

Comparative analysis of gene expression profiles among single cells is critically important to better understand the regulation of transcriptional activity, due to the substantial heterogeneity of transcriptional profiles across cell populations [1]. This issue is more pronounced in tissue samples and primary cells that involve multiple cell types or identities, rendering it indispensable to appropriately distinguish different cell types during analysis. Fluorescence in situ hybridization (FISH) [2–4] method has provided an avenue to investigate gene expressions in single cells. Single molecule FISH (smFISH) [5], with multiple fluorescent oligos

**Funding:** This work was supported by the National Institutes of Health (NIH) grants R01AG057497, R01EY027789 and UM1HG009402 to Y.S.

**Competing interests:** The authors have declared that no competing interests exist.

hybridized to each transcript, can quantitatively evaluate the stochastic expression of genes. RNAscope [6] has greatly improved the accuracy and efficacy in detecting the single molecule transcript by utilizing double Z probes and successive amplification. With the aforementioned methods, gene transcripts can be quantified by counting individual dots in 3D stacks at the single-cell level.

Various approaches have been developed to analyze data derived from FISH experiments [7–9]. These methods include a 3D Gaussian filtering step to correct illumination and a Laplacian of Gaussian enhancing step to obtain local maxima above a certain threshold [10]. Among them, *starfish* (https://spacetx-starfish.readthedocs.io/en/latest/), an open-source Python-based platform developed for analyzing spatial transcriptomics, is by far the most efficient imaging analysis tool in dealing with multiplexed spatial smFISH [11–17] and in-situ sequencing (ISS) data [18]. While *starfish* is useful for analyzing FISH data, it has several limitations. The first and foremost limitation is that *starfish* doesn't have the features for processing additional layers of information such as cell marker immunostaining in a heterogeneous cell population, hindering the analysis of gene transcription in complex developmental or pathological processes [19, 20]. Besides, there is still room for improvement in multiple aspects including signal to noise enhancement, precise nuclei segmentation, and options for adjusting parameters during intermediate steps, etc. Here, we present the Single-Molecule Automatic mRNA Transcription Quantification pipeline (SMART-Q) with flexible and user-friendly features to allow for automatic detection of gene transcript signals, immunofluorescence signals, and precise segmentation of single cells. SMART-Q can analyze multiple channels in a single pipeline, and can accurately and efficiently quantify cell type-specific single-molecule RNA through integration with cell markers with improved user experience.

## Materials and methods

### Cell culture

Tissues are dissected and primary cells are disassociated from developmental dorsal cortex according to the protocol from Nowakowski et al [21]. Samples were collected with prior informed consent in strict observance of legal and institutional ethical regulations. All protocols were approved by the Human Gamete, Embryo, and Stem Cell Research Committee (GESCR) and Institutional Review Board at the University of California, San Francisco. Cells were cultured on coverslips and infected with lenti-virus expressing either GFP or mCherry. Cells were fixed in 4% PFA on Day 4 for staining.

### RNAscope and immunocytochemistry staining

smFISH targeting nascent RNA of HES1 or BCL11A were performed using RNAscope® Multiplex Fluorescent Reagent Kit v2(ACDBio). Probes binding the intronic region of target genes were designed and synthesized by ACDBio. FISH signal was labeled with TSA Plus Cyanine 5 (Perkin Elmer). Immunocytochemistry was carried out after FISH procedure [22]. Antibodies targeting GFP(Abcam, ab1218), mCherry (Abcam, ab205402), GFAP (Ab4648) and SATB2 (Abcam, ab34735) were incubated overnight. Secondary antibodies including Alexa Fluor 594 Goat anti-chicken IgY secondary antibody (Thermo Fisher Scientific, A11042), Alexa Fluor 488 donkey anti-mouse IgG secondary antibody (Thermo Fisher Scientific, A21202), Alexa Fluor 546 donkey anti-mouse IgG secondary antibody (Thermo Fisher Scientific, A10036) and Alexa Fluor 488 donkey anti-rabbit IgG secondary antibody (Thermo Fisher Scientific, A21206) were incubated at RT for 1hr. Nuclei are stained with DAPI for 5 min before mounting with ProLong™ Gold Antifade Mountant (Thermo Fisher Scientific, P36930).

## Image acquisition

Images were acquired by TSC SP8 Leica equipped with a 40× 1.43 NA oil objective. 2 sequential scans were performed to avoid spectral overlap. The pixel size in the image plane is 0.285 μm ×0.285 μm. The Z-step size was 0.4μm.

## Code availability statement

The SMART-Q program is freely accessible on Github (https://github.com/shenlab-ucsf/SMART-Q).

## Results

### Enhanced architecture for source codes

In previous releases of *starfish*, the program was structured largely as a single executable function, requiring users to run through the entire pipeline before determining if a parameter choice is appropriate. In addition, when transfered to Jupyter Notebook, the default scripts written for Jupyter Notebook are not optimally organized or detailed for first-time users, creating an excessively front-loaded learning curve.

SMART-Q is created with a new coding architecture that simplifies and modularizes each step of the pipeline into modular architecture (Fig 1) with the option to change parameters and assess quality at each step of analysis. By standardizing each module of the pipeline, users can effortlessly flow through the pipeline and change parameters much more efficiently when needed without sacrificing accuracy and flexibility.

Specifically, we implement the workflow as follows: 3D stacks of images are converted into SMART-Q format for each experiment (Fig 1A and 1B). SMART-Q first filters images using Gaussian high pass and Gaussian low pass filters (Fig 1C(1)). A Gaussian high pass filters out background noise, while a Gaussian low pass amplifies and smooths signals from fluorescent spots [23]. The RNA signal is then detected in three dimensions by fitting Gaussians to fluorescent spots of the image (Fig 1C(2)) [10]. Segmentation is then performed on the nuclei channel in two dimensions to determine the location of each nucleus (Fig 1C(3a)). If nascent RNA is

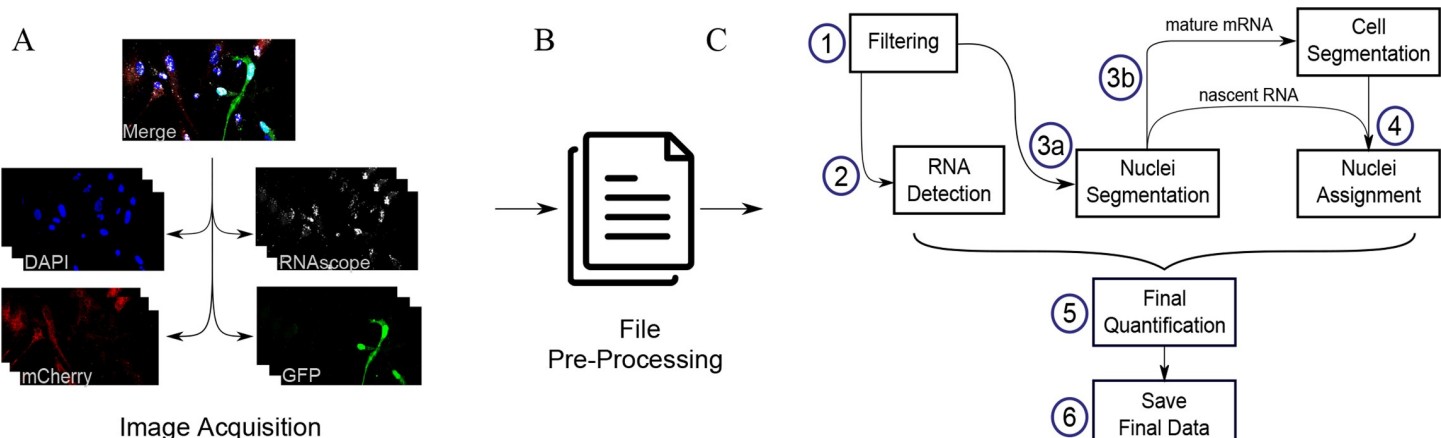

**Fig 1. Schematic of SMART-Q's workflow under the new coding architecture.** (A) 3D stacks of smFISH and Immunofluorescence images obtained by confocal. GFP and mCherry are stained to represent differrent cell types. (B) Image files are converted to SMART-Q format as input. (C) (1) *Filtering* removes noise and amplifies signals. (2) *Detection* finds all RNA transcripts. (3a) *Nuclei segmentation* identifies all nuclei in DAPI stain. (3b) If the user is quantifying mature mRNA, an additional step is implemented to determine coordinates of all positive cells in each channel. (4) Assign nuclei to cell type-specific channel(s). (5) Final images and (6) final data are saved as PNG and Excel.

the target of analysis, then nuclei are simply assigned to cell channel(s) (Fig 1C(4)). If mature mRNA is the target of analysis, then segmentation is also performed on the cell marker channels (Fig 1C(3b)), and then nuclei are automatically assigned to cell marker channel(s) (Fig 1C (4)). Finally, the positional data derived from RNA detection and segmentation are integrated to determine the final quantification of transcripts in each nucleus or cell (Fig 1C(5)).

At the end of the pipeline, additional features are added so that images are saved for a quick review of the results and optional quality assurance. The final results and metadata are saved in Excel and CSV format. Quantification results are saved in cumulative batch files for optimal analysis within Excel (Fig 1C(6)).

For users who wish to customize the pipeline by modifying or adding a step, the code has been optimized to make it easily readable and adaptable. Each channel type (transcripts, nuclei, cells) has been simplified to a Python class object, while each step of the pipeline is represented as a single function that belongs solely to the channel type(s) that uses it. With a specialized class for each of the three channel types, the code can easily accommodate any number of each channel type. In addition, with clutter reduced to an absolute minimum, users can efficiently and effortlessly locate relevant modules of the code that they wish to customize without having to waste time on irrelevant sections.

## Determination of the optimal threshold for RNA detection

The final detection of RNA transcripts heavily depends on the parameters chosen by the user. The parameter that has the most impact on results, called min_mass in the program, is the minimum intensity that a diffraction-limited spot must have in order to be recognized by the detection function. In the previous *starfish* pipeline, choosing the correct value for this parameter was a difficult task, as it was impossible to precisely compare the results of RNA detection

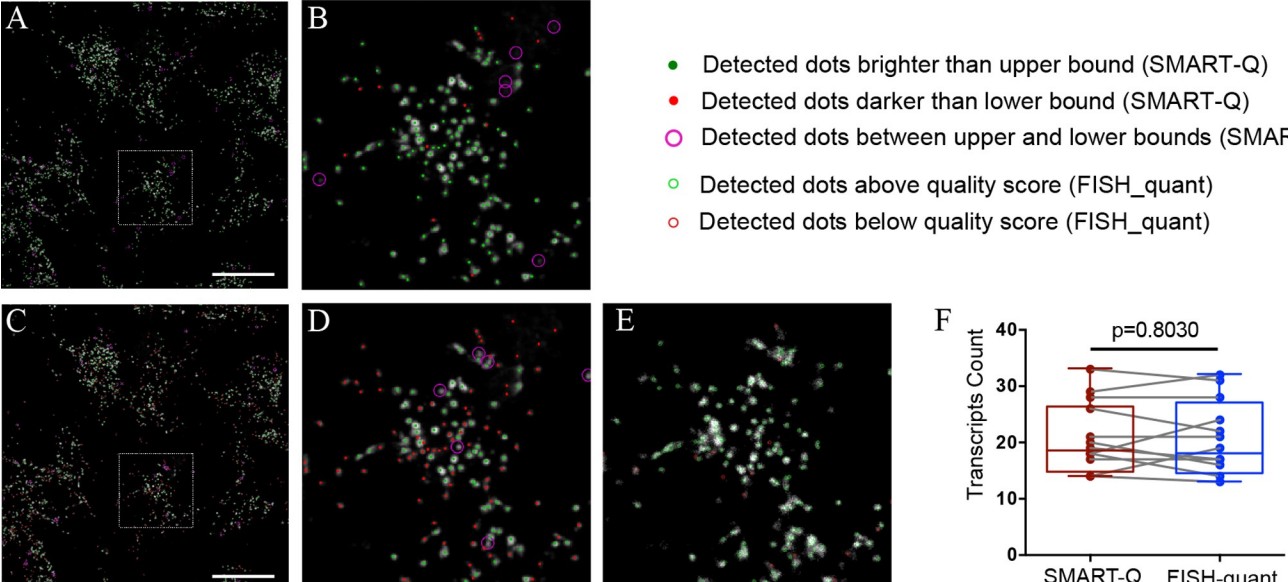

**Fig 2. Assessing the quality of RNA detection results.** Graphic representations of the intensities of all detected spots relative to a chosen upper bound intensity and lower bound intensity. By comparing the results of different bounds, the user can determine an optimal minimum brightness for detecting gene transcripts. (A, B) Upper bound intensity of 0.07 and lower bound of 0.06. (C, D) Upper bound of 0.24 and lower bound of 0.23. B and D are the zoom-in of boxed subregion in A and C. The scale bar in A and C are 50 μm. (E) RNA detection by FISH-quant with the same data source in B or D. Detected dots with min_intensity = 200, quality score>60 (green circles). (F) Paired RNA detection are performed among 5 coverslips using SMART-Q or RNAscope. Paired t test. N = 12, P = 0.8030.

to the image of the original RNA signal nor was it possible to quantify the intensity value of each detected spot. In order to solve this problem, our SMART-Q provides a visualization tool that overlays the results of detection onto the post-filtered image of the RNA signal. SMART-Q provides a default threshold for FISH signal detection, and additional higher and lower thresholds can be defined by users. The visualization utility can identify spots in each intensity interval, categorize them and overlay the detection results relative to an upper bound and lower bound of the user's choice (Fig 2A and 2C). Then users can quickly tell the quality of the spots identified within each interval (Fig 2B and 2D). The feature of displaying multiple intensity cutoffs simultaneously in SMART-Q drastically improve user's experience in choosing an appropriate value for the minimum intensity in RNA detection. Notabley, our method results in similar RNA counts compared to FISH-quant (Fig 2E and 2F).

## Increased accuracy and quality control for the segmentation of nuclei and cells

Segmentation has been recognized as a challenge shared by all existing methods. This is because segmentation results frequently have errors that must be corrected. Some methods, such as FISH-quant, use the Moore-Neighbor tracing algorithm modified by Jacob's stopping criteria in order to determine nuclei boundaries [24]. However, this approach requires users to manually trace every nucleus they wish to analyze, which is time-consuming and human labor intensive, and thus highly inefficient, especially for large data sets. Other methods, such as *starfish*, determine nuclei boundaries using the watershed algorithm, which finds nuclei boundaries based on local minima and produces a segmented region for each local minimum. However, this strategy is known for its tendency to over-segment cells, meaning that a single nucleus may be mistakenly fractured into multiple subcomponents [25–27]. Moreover, when nuclei or cells border each other, they are prone to be under-segmented, meaning that multiple nuclei may be merged as one. In addition, images may contain background noise or artifacts, which cannot be effectively removed by *starfish*. Thus, segmentation remains as one critical step in the pressing need for method improvement.

SMART-Q provides three solutions mitigating the aforementioned issues encountered in the segmentation. First, we have added a new parameter, called minimum depth to allieviate the over-segmentation potential that commonly occur in other analysis pipelines [25, 28]. When the watershed function classifies pixels by measuring saliency in contours, regions which can be taken as catchment basins are formed by local geometric structure. The minimum depth is a factor reflecting the height between watershed minima and various lower boundary points, or the height limits between neighboring catchment basins. Defining of the minimum depth enables sequential combination of watershed whose depth is below the minimum. With this additional parameter, the new watershed function in SMART-Q ensures that each local minima is significant enough to warrant separation of regions, thus effectively avoiding separations due to concavities with low depth. We recommend setting this factor at a value between $10^{-6}$ to $10^{-7}$, the optimal value for preventing over-segmentation based on our experience. On the other hand, watershed function with minimum depth provides solutions to handle cases of clustered nuclei or cells, which are more often observed in tissue sections. When bordering nuclei are under-segmented, depth in the bordering region tends to be shallow (Fig 3A), which shares similar curvature in over-segmented regions. Leveraging such tendency, the user may set the minimum depth to slightly higher values, which artificially over-segments the merged nuclei, thereby effectively separating nuclei that share a border and correcting the under-segmentation error. The optimal parameter we recommend is between $5^{*}10^{-5}$ and $5^{*}10^{-6}$ for mammalian cells (Fig 3B).

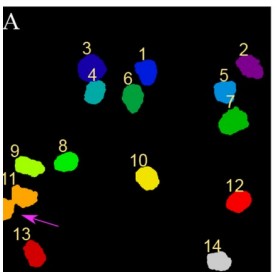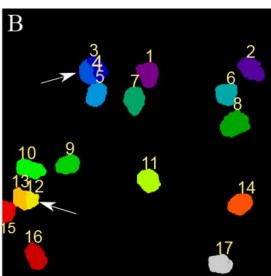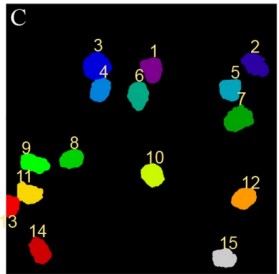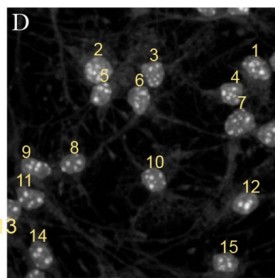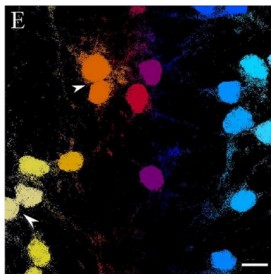

**Fig 3. Use of minimum depth and correction in segmentation.** (A) Segmentation of nuclei is performed with *starfish*'s default settings. Magenta arrow designates undersegmentation, or multiple nuclei erroneously merged together. No minimum depth was used. (B) Minimum depth of $10^{-5}$ is used to artificially oversegment the previously undersegmented nuclei. White arrow designates oversegmented nuclei. (C) Oversegmentation can easily be corrected using the new segmentation correcting function. (D) The original nuclei image, which accurately resembles the final nuclei segmentation. (E) Outline of nuclear by FISH-quant. Entire image was seleted for autodetection. Empty arrows are pointing to the under-segmented nuclear that need to be separated manually. The scale bar is 10 μm.

Second, SMART-Q enables an additional parameter: a minimum size parameter. Use of this minimum size parameter allows for automatic removal of artifacts and background noise. Because dead cells' nuclei diminish in size upon death, they can easily be removed during the process of segmentation with the use of the minimum size parameter.

Finally, for nuclei that have been mis-identified or over-segmented that cannot be fixed by parameter adjustment, SMART-Q provides a new function for direct quality correction. Previously in *starfish*, the user was unable to fix the results of segmentation directly and instead had to rely on iterative alteration of the parameters to achieve desirable results. Our method, in contrast, first visualizes the results of segmentation, giving a unique ID to each segmented area. With these unique IDs, users are empowered to perform flexible manipulation of the segmentation results, such as complete deletion of certain region(s) or merging multiple into one, by simply feeding SMART-Q ID(s) of the corresponding region(s) (Fig 3B, 3C and 3D), thus providing the feasibility of manual correction of incorrectly assigned nuclei after the visual inspection and verification. When the same image is analyzed by outline function in FISH-quant (Fig 3E). The entire imgae is selected and nuclear are automatically detected by intensity threshold that can't be adjusted. When the FISH-quant failed to separated two closeby nucleus, one have manually dividing two joint nucleus by drawing the boundary which can be very inefficient with the risk of introducing bias.

The implementation of new parameters minimum depth and minimum size significantly improves the accuracy and flexibility of the segmentation step. Any remaining issues in segmentation can now be directly fixed using our new quality correction function, substantially reducing the amount of time and effort to achieve accurate segmentation.

## Addition of a new feature to enable assignment of nuclei to cell types

RNA quantification experiments involving heterogeneous populations often seek to understand the differences between the various cell types or identities analyzed. For example, a researcher may aim to identify the differences in gene expression between wild type cells and CRISPR-modified cells or between differentiated and undifferentiated cells. In order to detect these differences, one must be able to accurately and efficiently assign identity to each nucleus or cell to achieve precise quantification of RNA transcripts in a cell type-specific manner.

To maximally benefit from this new functionality, we need to determine both positional data of each nucleus (e.g., DAPI staining) and each cellular channel (e.g., IF staining of cell markers). Here we infected radial glia cells with GFP and mCherry expressing lenti-vrius and followed by and performed immunocytochemistry of GFP and mCherry proteins to model

different cell types and FISH staining targeting intronic region of HES1 as an example. In SMART-Q, we determine the positional data of each nucleus by performing segmentation on the DAPI staining (Fig 4A and 4E). Cell staining is categorized using immunofluorescence properties that differ across cell types, such that cells will only exhibit positive staining if they belong to a particular cell type or identity according to the user's experimental design (Fig 4C and 4D). When quantifying the nascent RNA and the mature mRNA, different strategies are employed to determine the positions of channel-positive cells. In the case of mature mRNA quantification where RNA signals are distributed in both nucleus and cytoplasm, the user is required to perform segmentation on both cell identity and morphology channel. Therefore, we are able to integrate the data derived from cellular segmentation to determine which nuclei belong to channel-positive cells, and outline the territory for counting FISH signals. In current version of SMART-Q, cell segmentation is based on 2D segmentation algorithm and might not accurate when there're overlapping boundaries. So we are only supporting cell segmentation based on DAPI stating. In the case of nascent RNA quantification (Fig 4B), where cellular segmentation is not performed, we implement an approach that requires merely a threshold to determine the position of each bright region in a cell staining. The data is then integrated with nuclei segmentation results to automatically assign nuclei to channel-positive cells (Fig 4G and 4H). In both cases, users can perform quality control to ensure correct results. Next we validated cell type specific analysis in a human primary cells from the developing cortex. Here radial glia cells (RG) are labeled with GFAP antibody and excitatory neurons (eNs) are labeled

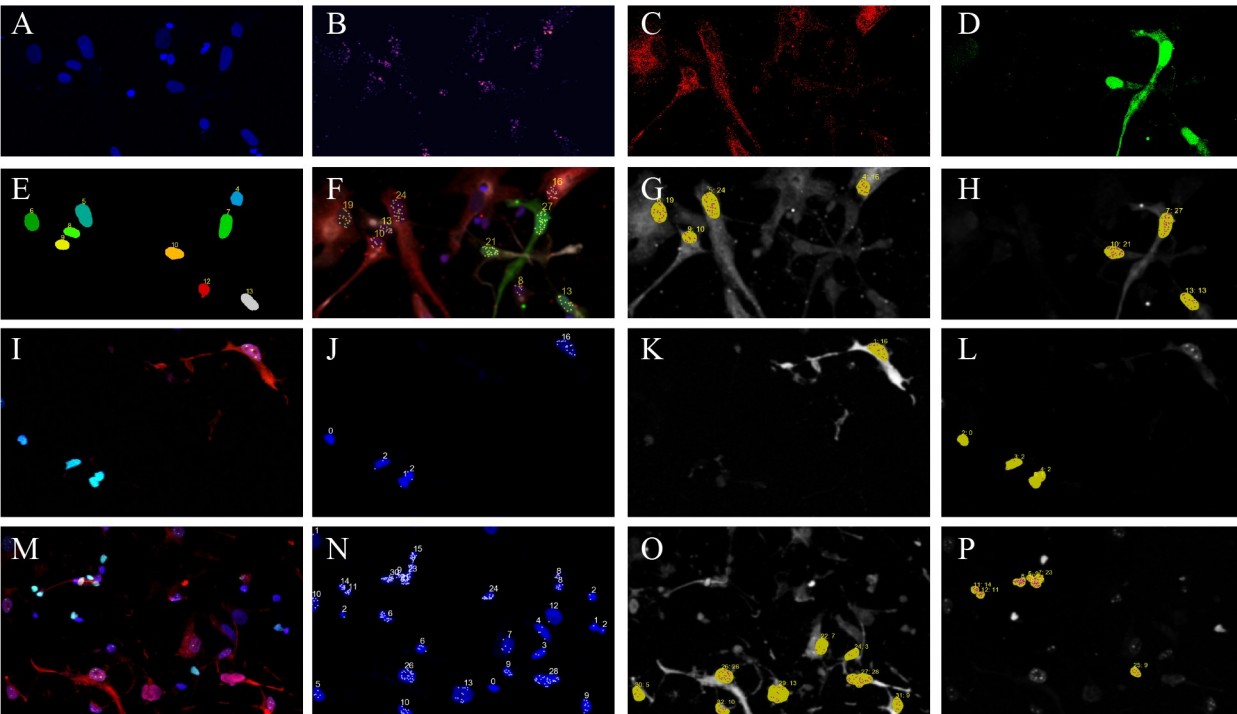

**Fig 4. Integrative analysis of cell type-specific transcirpt counts by SMART-Q.** (A, C, D) Immunocytochemistry staining for nuclear, GFP and mCherry. (B) RNAscope targeting HES1 transcripts in radial glia cells. (E) Nuclear segmentation. (F) Pseudocolor composition of all channels, including cell type-specific transcript counting. (G, H) Transcript counts in GFP/mCherry positive cells. (I, M) Composite image of immunocytochemistry staining in primary cells. Radial glia cells are labeld with GFAP (red), and excitatory neurons are labeled with SATB2 (green). (J,N) Nascent RNA transcript counts in all nuclei. (K, O) Nascent RNA transcript counts in identified radial glia cells. (L, P) Nascent RNA counts in identified excitatory neurons. (I-L) RNAscope with probes targeting intronic region of *HES1*. (M-P) RNAscope with probes targeting intronic regions of *BCL11A*. The scale bars are 30 μm.

with SATB2 antibody on the same slide. We quantified the expression of *HES1* gene that are only expressed in RG and *BCL11A* gene expressed in both RG and eNs. We demonstate cell type-specific expression of *HES1* in GFAP positive cells (Fig 4I–4L), while *BCL11A* are expressed in both cell types (Fig 4M–4P) by RNAscope.

## Streamlined procedure for saving data and parameter settings

While developing SMART-Q, we have identified a need for saving data for subsequent analyses in other programs, which might be implemented in platforms other than Python, such as R, Excel, or Google Sheets. In order to expedite this process, we chose to save final results in a batch file containing the results for every sample analyzed in a batch. We have written template files for post-analysis in Google Sheets, which are available to users to customize for their purposes.

We have also added a new feature that allows users to save settings and metadata for re-establishing the analysis of a specific sample. Previously in *starfish*, once analysis was complete, the settings for the previous sample were not stored. If a change needed to be made to a parameter or step in the pipeline in the future, it was impossible to re-establish the pipeline with the parameters previously used. In SMART-Q, a metadata file with all parameters and settings is now saved each time during the analysis. These metadata files are not saved in batch as final results are, but are rather saved in individual files for each sample analyzed. If a parameter or step of the pipeline is later desired to be altered, this metadata file can be used as input in a modified version of the pipeline to re-establish the pipeline with the previous parameters, while still allowing the user to change any parameters or steps of their choice.

## Simplified image file input feature and streamlined visualization for results display and quality assurance

We further empower the SMART-Q with the ability to execute ImageJ or Fiji macro scripts, a feature that was lacking in other analysis pipelines. Users can now create composite images of the final results and execute any ImageJ and Fiji macros within SMART-Q, making it possible to create desired images in a batch analysis, drastically reducing the amount of time this process would otherwise require (Fig 4F).

We additionally find it useful to save images at each step of the pipeline for quality assurance. We designed images that best exemplify the effects of each step and now display and save images for the following steps: filtering, RNA detection, segmentation, channel assignment, and final quantification. These will allow users to review critical quality checkpoints whenever desired without having to re-run the pipeline.

## Discussion

SMART-Q is time-saving for analyzing large datasets. For one image with 4 channels and 20 Z-stacks, it takes about 5 min from confocal image output to cell type-specific RNA counts. In comparison to FISH-quant, we found FISH-quant spends similar amount of time in splitting images and 3D dot identification, but much more time in segmentation when manually outlining multiple nuclei. Notably, multiple jupyter-notebook interfaces can run simultaneously, while other pipelines could only analyze images one by one. Thus, by using SMART-Q, users can save days of time when processing hundreds of images.

Some improvements can be implemented to make SMART-Q even more powerful in cell identity assignment in the future. Currently, segmentation is performed on the 2D axis rather than the 3D axis, which creates issues when a significant portion of multiple nuclei or cells share the same z plane. In highly confluent tissue samples, the lack of an adequate method to

accurately segment overlapping nuclei or cells suggests that the RNA transcripts belonging to overlapping regions cannot be properly quantified, and thus they must be removed from analysis during quality control. Similarly, overlapping nuclei can be difficult to assign to a particular cell type, as it may be difficult to determine which nucleus belongs to a channel-positive cell. Future efforts are warranted to ameliorate this issue by expanding the capability to segment along the z-axis as well or by integrating an optional hand-drawing or semi-supervised drawing application into the segmentation method.

## Conclusion

We have developed SMART-Q to quantify RNA transcripts at the single cell level with assigned cellular identity. Through its new modular design and a strong focus on ease of use and customizability, SMART-Q is applicable to any 3D FISH images, solving cell type-specific transcripts quantification in different experimental approaches. SMART-Q can automatically assign nuclei to cell channels, allowing users to compare results, both quantitatively and visually, among any number of cell types depending on the user's experimental design. SMART-Q provides quality control functionalities to test varying thresholds for RNA transcripts detection, and to improve nuclei and cell segmentation. Finally, by saving both quantitative and qualitative results, SMART-Q enhances users' capabilities with respect to quality assurance and streamlined analysis of quantification results. Overall, the streamlined and modular characteristics of SMART-Q significantly improve the user experience and makes cell type-specific RNA quantification analysis highly accurate and efficient.

## Acknowledgments

We thank the Shen lab members for providing critical feedback on the manuscript and the analysis pipeline. We thank *starfish* for providing the initial code applied in our pipeline.

## Author Contributions

**Conceptualization:** Xiaoyu Yang, Yin Shen.

**Data curation:** Xiaoyu Yang, Lenka Maliskova.

**Formal analysis:** Xiaoyu Yang.

**Funding acquisition:** Yin Shen.

**Investigation:** Xiaoyu Yang.

**Methodology:** Xiaoyu Yang.

**Project administration:** Xiaoyu Yang.

**Resources:** Mark-Phillip Pebworth, Arnold R. Kriegstein.

**Software:** Seth Bergenholtz.

**Supervision:** Xiaoyu Yang, Yin Shen.

**Validation:** Xiaoyu Yang.

**Visualization:** Xiaoyu Yang.

**Writing – original draft:** Seth Bergenholtz.

**Writing – review & editing:** Xiaoyu Yang, Yun Li, Yin Shen.

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
