## [Decision Letter · Decision Letter 0]

7 Feb 2020

PONE-D-20-01686

SMART-Q:  An Integrative Pipeline Quantifying Cell Type-Specific RNA Transcription

PLOS ONE

Dear Dr. Shen,

Thank you for submitting your manuscript to PLOS ONE. After careful consideration, we feel that it has merit but does not fully meet PLOS ONE’s publication criteria as it currently stands. Therefore, we invite you to submit a revised version of the manuscript that addresses the points raised during the review process.

Reviewers are suggesting to test SMART-Q for more than one RNA and cell types, and provide the demonstration of the segmentation of both of nulcei and cells.

We would appreciate receiving your revised manuscript by Mar 23 2020 11:59PM. To enhance the reproducibility of your results, we recommend that if applicable you deposit your laboratory protocols in protocols.io, where a protocol can be assigned its own identifier (DOI) such that it can be cited independently in the future. For instructions see: http://journals.plos.org/plosone/s/submission-guidelines#loc-laboratory-protocols

We look forward to receiving your revised manuscript.

Kind regards,

Ruijie Deng

Academic Editor

PLOS ONE

Journal Requirements:

2. In your Methods, please state the exact origin of the cells used in your study.

3. Please amend the manuscript submission data (via Edit Submission) to include author Lenka Maliskova.

Reviewers' comments:

Reviewer's Responses to Questions

**Comments to the Author**

1. Is the manuscript technically sound, and do the data support the conclusions?

Reviewer #1: Yes

Reviewer #2: Partly

2. Has the statistical analysis been performed appropriately and rigorously? 

Reviewer #1: Yes

Reviewer #2: Yes

3. Have the authors made all data underlying the findings in their manuscript fully available?

Reviewer #1: Yes

Reviewer #2: Yes

4. Is the manuscript presented in an intelligible fashion and written in standard English?

Reviewer #1: Yes

Reviewer #2: Yes

5. Review Comments to the Author

Reviewer #1: In this manuscript, the author developed an efficient and customizable analysis method, which can analyze gene transcripts in a cell type-specific manner. The method is facile and bring great convenience for users. I recommend its publication after some revisions as below:

1. The method is based on Z-stack images, RNA amplicons may merged together, or when target RNA has high expression, many amplicons may be too close to each other to be distinguished. The accuracy for RNA detection may be limited.

2. The author stated that their method has increased accuracy and quality control for the segmentation of nuclei and cells. However, they just showed the data for the segmentation of nuclei, while didn’t show the evidence of the segmentation of cells. Clinging cells always exist, and cell boundaries is more difficult to determined.

3. Another issue is how about the processing time SMART-Q required.

Reviewer #2: In this manuscript, the authors develop an efficient and customizable method to analyze data derived from FISH or RNAscope experiments, called Single-Molecule Automatic RNA Transcription Quantification (SMART-Q). The SMART-Q improves the features for processing additional layers of information compared with starfish, which is an open-source Python-based platform. It efficiently infers cell identity information from multiplexed immuno-staining and quantifies different cells with assigned cellular identity using a 3D Gaussian fitting algorithm. Furthermore, the authors have optimized SMART-Q for user experiences, such as flexible parameters specification, batch data outputs, and visualization of analysis results. SMART-Q may meet the demands for efficient quantification of single-molecule RNA and can be widely used for cell type-specific RNA transcript analysis.

I think this work would meet the criteria of PLOS ONE, if the following issues are properly addressed.

1. The author claimed that the SMART-Q can quantify cell type-specific RNA transcription. However, for cell type, the authors use the same type of cell infected with lenti-virus expressing either GFP or mCherry to character different cellular identity, this is not strict and accurate; for specific RNA, the authors only use a nascent RNA of HES1, which can't reflect cell type-specific RNA transcription. Therefore, I suggest that two different cell types and two cell type-specific RNA are needed in this work.

2. After infected with lenti-virus expressing either GFP or mCherry, cells have already produced different fluorescence signals, which could be effectively distinguished. But, why was still immunocytochemistry carried out targeting GFP and mCherry? Please explain it.

3. Most of the references are published before 5-10 years, and it is recommended to update the references.

6. PLOS authors have the option to publish the peer review history of their article (what does this mean?). If published, this will include your full peer review and any attached files.

Reviewer #1: No

Reviewer #2: No

---

## [Author Response · Author response to Decision Letter 0]

22 Mar 2020

Journal Requirements:

Thanks for editors comments. We checked the style and edited accordingly. 

2. In your Methods, please state the exact origin of the cells used in your study.

Cell source are updated with more details in method. Line 97-103

3. Please amend the manuscript submission data (via Edit Submission) to include author Lenka Maliskova.

Author list is amended to include Lenka Maliskova. We have also added Mark-Phillip Perbworth and Arnold Kriegstein due to their contributions for the manuscript revision.

Review Comments to the Author:

Reviewer #1: 

In this manuscript, the author developed an efficient and customizable analysis method, which can analyze gene transcripts in a cell type-specific manner. The method is facile and bring great convenience for users. I recommend its publication after some revisions as below:

1. The method is based on Z-stack images, RNA amplicons may merged together, or when target RNA has high expression, many amplicons may be too close to each other to be distinguished. The accuracy for RNA detection may be limited.

We appreciate reviewer#1’s recognition of the efficiency of SMART-Q in cell type-specific analysis of the FISH signal. When imaging RNAscope slides, we found the average size of RNA amplicons are ~800nm. Each dot exists in about 3~4 Z-stacks. If most of the dots are merged, such as mean diameter is more than 3~5um, FISH methods need to be adjusted to get more specific and clear staining. If there’s only a small proportion of merged amplicon of regular size, it can be corrected in SMART-Q. In the 3D Gaussian algorithm, the fluorescent intensity is fitting into Gaussian distribution both in XY and XZ direction. In this case, overlapping dots in XY direction can still be separated by the saddle between the two points in XZ direction. Besides, in fluorescent image analysis, 3D imaging is more comprehensive and accurate than 2D in FISH quantification as the number of dots might be different in different layers of the Z-axis.

2. The author stated that their method has increased accuracy and quality control for the segmentation of nuclei and cells. However, they just showed the data for the segmentation of nuclei, while didn’t show the evidence of the segmentation of cells. Clinging cells always exist, and cell boundaries is more difficult to determined.

SMART-Q can do 2D cell segmentation, which bases on the same algorithm as nuclear segmentation. However, the current SMART-Q version focuses on the nascent RNA quantification, and cell segmentation is not the first segmentation choice. We agree with the difficulties Reviewer#1 mentioned in identifying cell boundaries. 3D segmentation could perhaps better solve the problem but need perfect cell morphology staining and lots of computing power, which will compromise the efficiency of SMART-Q aimed in this version. We adjusted the description not to overclaim the function (line 269-271) and will incorporate an updated 3D segmentation algorithm into SMART-Q in the future to serve more applications. 

3. Another issue is how about the processing time SMART-Q required.

SMART-Q is timesaving when analyzing large datasets. For one image with four channels and 20 Z-stacks, it takes about 1 min to split an image into individual channels and stacks, 1 min to format into SMART-Q inputs and 3 min to perform 3D analysis in jupyter-notebook. We compared the processing time required with FISH-quant. FISH-quant spends a similar amount of time in splitting images and 3D dot identification, but much more time in segmentation due to outlining cells or nuclei manually. Notably, multiple jupyter-notebook interfaces can run simultaneously, while other pipelines could only analyze images one by one. When processing hundreds of images, this will save days. We add these descriptions to the manuscript line 327-363.

Reviewer #2: In this manuscript, the authors develop an efficient and customizable method to analyze data derived from FISH or RNAscope experiments, called Single-Molecule Automatic RNA Transcription Quantification (SMART-Q). The SMART-Q improves the features for processing additional layers of information compared with starfish, which is an open-source Python-based platform. It efficiently infers cell identity information from multiplexed immuno-staining and quantifies different cells with assigned cellular identity using a 3D Gaussian fitting algorithm. Furthermore, the authors have optimized SMART-Q for user experiences, such as flexible parameters specification, batch data outputs, and visualization of analysis results. SMART-Q may meet the demands for efficient quantification of single-molecule RNA and can be widely used for cell type-specific RNA transcript analysis.

I think this work would meet the criteria of PLOS ONE, if the following issues are properly addressed.

1. The author claimed that the SMART-Q can quantify cell type-specific RNA transcription. However, for cell type, the authors use the same type of cell infected with lenti-virus expressing either GFP or mCherry to character different cellular identity, this is not strict and accurate; for specific RNA, the authors only use a nascent RNA of HES1, which can't reflect cell type-specific RNA transcription. Therefore, I suggest that two different cell types and two cell type-specific RNA are needed in this work.

We appreciate Reviewer 2’s comments. To adequately address the concern, we used the primary culture of the human embryonic cortex, which had a mixture of radial glia (RG), intermediate progenitor cells (IPC), excitatory neurons (eN) and inhibitory neurons (iN). As a proof of principle, we picked an RG-specific gene, HES1, with its expression in GFAP (a marker for RG) positive cells, but not in SATB2 positive cells that are eN (Fig4. I-L). We also picked another gene BCL11A, which is expressed in both RG and eN. We show the detection of positive FISH signals in both cell types using BCL11A FISH probes (Fig4. M-P). These results demonstrate the capability of SMART-Q in cell-type-specific transcription analysis. (Line 276-292)

2. After infected with lenti-virus expressing either GFP or mCherry, cells have already produced different fluorescence signals, which could be effectively distinguished. But, why was still immunocytochemistry carried out targeting GFP and mCherry? Please explain it.

For the lenti-virus vector induced expression of GFP and mCherry, we could see the fluorescence in live cells, but it will be bleached by RNAscope staining procedure. To overcome this issue, we had to use the immunocytochemistry to enhance the signal.

3. Most of the references are published before 5-10 years, and it is recommended to update the references.

Thanks for Reviewer #2’s suggestion. Some newly published references, including ref 4, 7, 21, 22 are added.

---

## [Decision Letter · Decision Letter 1]

1 Apr 2020

SMART-Q:  An Integrative Pipeline Quantifying Cell Type-Specific RNA Transcription

PONE-D-20-01686R1

Dear Dr. Shen,

We are pleased to inform you that your manuscript has been judged scientifically suitable for publication and will be formally accepted for publication once it complies with all outstanding technical requirements.

With kind regards,

Ruijie Deng

Academic Editor

PLOS ONE

Additional Editor Comments (optional):

Reviewers' comments:

Reviewer's Responses to Questions

**Comments to the Author**

1. If the authors have adequately addressed your comments raised in a previous round of review and you feel that this manuscript is now acceptable for publication, you may indicate that here to bypass the “Comments to the Author” section, enter your conflict of interest statement in the “Confidential to Editor” section, and submit your "Accept" recommendation.

Reviewer #1: All comments have been addressed

Reviewer #2: All comments have been addressed

2. Is the manuscript technically sound, and do the data support the conclusions?

Reviewer #1: Yes

Reviewer #2: Yes

3. Has the statistical analysis been performed appropriately and rigorously? 

Reviewer #1: Yes

Reviewer #2: Yes

4. Have the authors made all data underlying the findings in their manuscript fully available?

Reviewer #1: Yes

Reviewer #2: Yes

5. Is the manuscript presented in an intelligible fashion and written in standard English?

Reviewer #1: Yes

Reviewer #2: Yes

6. Review Comments to the Author

Reviewer #1: (No Response)

Reviewer #2: (No Response)

7. PLOS authors have the option to publish the peer review history of their article (what does this mean?). If published, this will include your full peer review and any attached files.

Reviewer #1: No

Reviewer #2: No

---

## [Editor Report · Acceptance letter]

17 Apr 2020

PONE-D-20-01686R1 

SMART-Q:  An Integrative Pipeline Quantifying Cell Type-Specific RNA Transcription 

Dear Dr. Shen:

I am pleased to inform you that your manuscript has been deemed suitable for publication in PLOS ONE. Congratulations! Your manuscript is now with our production department. 

With kind regards,

on behalf of

Dr. Ruijie Deng 

Academic Editor

PLOS ONE